# Genetic Involvement of *Mycobacterium avium* Complex in the Regulation and Manipulation of Innate Immune Functions of Host Cells

**DOI:** 10.3390/ijms22063011

**Published:** 2021-03-16

**Authors:** Min-Kyoung Shin, Sung Jae Shin

**Affiliations:** 1Department of Microbiology and Convergence Medical Sciences, Institute of Health Sciences, College of Medicine, Gyeongsang National University, Jinju 52727, Korea; mkshin@gnu.ac.kr; 2Department of Microbiology and Institute for Immunology and Immunological Diseases, Brain Korea 21 Project for Medical Science, Yonsei University College of Medicine, Seoul 03722, Korea

**Keywords:** *Mycobacterium avium* complex (MAC), *M. avium* subsp. *hominissuis*, virulence-associated genes, epithelial cells, macrophages

## Abstract

*Mycobacterium avium* complex (MAC), a collection of mycobacterial species representing nontuberculous mycobacteria, are characterized as ubiquitous and opportunistic pathogens. The incidence and prevalence of infectious diseases caused by MAC have been emerging globally due to complications in the treatment of MAC-pulmonary disease (PD) in humans and the lack of understating individual differences in genetic traits and pathogenesis of MAC species or subspecies. Despite genetically close one to another, mycobacteria species belonging to the MAC cause diseases to different host range along with a distinct spectrum of disease. In addition, unlike *Mycobacterium tuberculosis*, the underlying mechanisms for the pathogenesis of MAC infection from environmental sources of infection to their survival strategies within host cells have not been fully elucidated. In this review, we highlight unique genetic and genotypic differences in MAC species and the virulence factors conferring the ability to MAC for the tactics evading innate immune attacks of host cells based on the recent advances in genetic analysis by exemplifying *M. avium* subsp. *hominissuis*, a major representative pathogen causing MAC-PD in humans. Further understanding of the genetic link between host and MAC may contribute to enhance host anti-MAC immunity, but also provide novel therapeutic approaches targeting the pangenesis-associated genes of MAC.

## 1. Introduction

Nontuberculous mycobacteria (NTM), a ubiquitous opportunistic pathogen, has become a major public health problem worldwide in recent decades, following a significant increase in its incidence and prevalence in humans [1,2]. The distribution of NTM species that cause pulmonary disease (PD) differs according to countries and regions, although *Mycobacterium avium* complex (MAC) is the most common cause worldwide [3]. MAC lung disease is divided into two main types of disease: the nodular bronchiectatic and fibrocavitary forms, and the proportion of cases classified as the nodular bronchiectatic form is increasing significantly [4,5]. In particular, in Korea, it was reported that the incidence rate of NTM-PD increased with the increase of MAC-PD, while the incidence of PD was maintained in NTM strains excluding MAC [5].

It is known that NTM infection, including MAC, is not transmitted human to human, instead being acquired from the environment [6]. MAC organisms inhabit non- biological or biological resources such as soil, water, food, and animals, and can cause various types of disease in humans, other mammals and birds [6,7]. These bacteria are also known to have the ability to survive in a wide range of environmental conditions, including low pH, extreme temperatures, low oxygen levels, and the presence of chlorine or ozone [6]. Therefore, as the habitat spectrum is wide, it is believed that various survival strategies can be used to survive within the host [6,8]. Although MAC organisms are closely genetically related, they exhibit different host susceptibilities and cause different disease types [6,9]. In addition, the MAC genes that play a role in etiology, drug resistance, and immune regulatory mechanisms in host target cells have not been fully identified. As mentioned above, it is believed that MAC organisms choose survival strategies according to their respective circumstances, and have evolved accordingly. Therefore, in this review, we intend to describe the differences in the genetic characteristics of MAC organisms and explain the role of virulence genes on evasion of the host’s innate immunity, which provides insight into the survival strategy of MAC in the host.

## 2. Comparative Genomics of MAC

MAC is the most frequently isolated species of NTM worldwide causing human disease [3]. New MAC species and subspecies are constantly being identified as the understanding of genetic diversity, differential pathogenicity, and various infectious agents of MAC is increasing [6]. MAC traditionally includes two major species; *M. intracellulare* and *M. avium* [3,10]. The latter is the most clinically important species in humans and animals, and consists of four subspecies, namely *M. avium* subsp. *avium* (MAA), *M. avium* subsp. *silvaticum* (MAS), *M. avium* subsp. *paratuberculosis* (MAP), and *M. avium* subsp. *hominissuis* (MAH), each of which have specific pathogenic and host range characteristics [6,11,12]. MAC has different virulence and ecology among subspecies; some species, including MAA, MAP and MAS, are stringent pathogens, while *M. intracellulare* and MAH are considered to be widely distributed environmental bacteria [6,11]. Although these environmental bacteria traditionally believed to be non-pathogenic, they can cause severe types of disease along with destructive tissue lesions even in immunocompetent individuals [12]. Moreover, different MAC organisms each exhibit definite host specificity, disease type, and epidemiologic characteristics, and many researchers have tried to explain this phenomenon by analyzing the genome contents of the species [13]. Therefore, we intend to understand the genetic characteristics of MAC organisms and explain the range of genetic determinants for host specificity and pathogenicity, including genetic evolution.

### 2.1. Recent Advance in Classification and Identification of MAC Organisms

Recent advances in molecular analysis have made it possible to identify and classify new (sub)species within the MAC at the molecular level [6]. In addition to the two species of *M. avium* and *M. intracellulare* in MAC, *M. chimaera* [14], *M. colombiense* [11], *M. arosiense* [15], *M. vulneris* [16], *M. bouchedurhonense* [17], *M. marseillense* [17], *M. timonense* [17], *M. paraintracellulare* [18], newly defined *M. intracellulare* subspecies, including *M. intracellulare* subsp. *yongonense* [19] and *M. indicus* pranii [20], have also been reported [6]. Table 1 summarizes the known information and genetic characteristics of these newly identified MAC members.

MAA is a strain of the genotypes IS*901*+ and IS*1245*+ and serotypes 1, 2, and 3 in the MAC isolates [21] (Table 1). Prior to being determined as MAA, the bacterium was simply called *M. avium* and has long been recognized as a major pathogen in avian tuberculosis in wild and domestic birds [11,22,23]. MAA has been reported to cause pulmonary infection in adults, adenopathy in children, and disseminated infection in patients with acquired immunodeficiency syndrome [11,24,25]. Although MAA has previously been isolated from a variety of animal species, including cattle, sheep, goats, pigs, cats, kangaroos, and humans [22,26], it is considered to be highly host-specific and a strict pathogen to birds. MAS, like MAA, is an avian pathogen and was first denoted as “wood pigeon *Mycobacterium*” before being classified as MAC in 1990 [26] (Table 1). In addition to wood pigeons [23,26], MAS has been reported separately in cranes [26], penguins [24], roe deer [25], and hazel hens [25]. MAS, like MAA, has both the IS*1245* and IS*901* genotypes, such that MAS can be identified by phenotypic characteristics alone, including mycobactin-dependent growth, growth stimulation at pH 5.5, and growth inhibition in egg media [26,27]. Mycobactin dependence is known to disappear after primary culture, the phenotypes of which can be mutated or erroneously observed, potentially exhibiting different properties for isolates within the same subspecies [26,27].

**Table 1 ijms-22-03011-t001:** Genetic makers and clinical features of the representative strains in MAC (sub)species.

Species	Strain	Host	Isolate Origin	Genetic Features		Clinical Features	Year of First Description	References
				IS *	ITS1 **			
I. *Mycobacterium avium* subspecies								
*Mycobacterium**avium* subsp. *avium*	ATCC25291	Chicken	Denmark	IS*901* + IS*1245* +	Mav-A	Avian tuberculosis	1990	[26]
*Mycobacterium**avium* subsp. *hominissuis*	IWGMT49	Pig	Netherlands	IS*901* – IS*1245* +	Mav-A	Pulmonary disease	2002	[21]
*Mycobacterium**avium* subsp. *paratuberculosis*	ATCC19698	Cow	USA	IS*900* +	Mav-A	Johne’s disease	1990	[26]
*Mycobacterium avium* subsp. *silvaticum*	ATCC49884	Wood pigeon	France	IS*901* + IS*1245* +	Mav-A	tuberculosis in birds and paratuberculosis in mammals	1990	[26]
II. Species/subspecies closely related to *Mycobacterium intracellulare*								
*Mycobacterium* *intracellulare*	ATCC15985	Human	-	-	Min-A	Pulmonary disease	1965	[10]
*Mycobacterium chimaera*	DSM44623 CIP 107892	Human	Italy	IS*900* – IS*901* – IS*1245* -	MAC-A	Pulmonary disease	2004	[14]
*Mycobacterium colombiense*	CIP108962	Human	Colombia		MAC-X	Bacteremia; lymphadenopathy	2006	[11]
*Mycobacterium arosiense*	DSM45069	Human	-	-	-	Disseminated osteomyelitis in immunocompromised child	2008	[15]
*Mycobacterium vulneris*	DSM45247 CIP 109859	Human	Netherlands		MAC-Q	A suppurative wound consequent to a dog bite/ cervical lymphadenitis in a child	2009	[16]
*Mycobacterium bouchedurhonense*	CIP109827	Human	France	-	-	Pulmonary disease	2009	[17]
*Mycobacterium marseillense*	CIP109828	Human	France	-	-	Pulmonary disease with bilateral bronchiectasis and multiple nodule	2009	[17]
*Mycobacterium timonense*	CIP109830	Human	France	-	MAC-K	Pulmonary disease	2009	[17]
*Mycobacterium* *paraintracellulare*	KCTC 29084	Human	Korea	-	MIN-A	Pulmonary disease	2016	[18]
*Mycobacterium intracellulare* subsp. *yongonense*	DSM45126	Human	Korea	-	-	Pulmonary disease	2013	[19]
*Mycobacterium indicus pranii*	MTCC 9506 DSM 45239	-	India	-	-	Leprosy vaccine candidates (previously referred to Mycobacterium ‘w’)	2008	[20]

***** IS, Insertion Sequence ** ITS1, the internal transcribed spacer 1 region of rRNA genes.

MAP causes Johne’s disease or paratuberculosis, resulting in chronic granulomatous enteritis in ruminants and wildlife [28], causing chronic diarrhea, weight loss, and eventual death in animals; this can cause significant economic losses for farms infected by this mycobacterium [28]. For these reasons, MAP is the most important MAC member in the field of veterinary medicine, although it can also cause infections and diseases in other animal species, including non-human primates [29]. More importantly, MAP has been found to be associated with Crohn’s disease in humans [29,30]. MAH isolates were designated [31,32] as MAH, which is less virulent for birds, but more frequently isolated from humans and pigs. It is considered ubiquitous in the environment and can cause serious disseminated infections in immunocompromised patients, such as those infected with HIV [31,32]. In addition, MAH has been demonstrated to cause cervical lymphadenitis in children with cystic fibrosis and pulmonary infections, and even as an opportunistic pathogen in immunocompetent humans [12,32,33]. *M. intracellulare* was first described as *Nocardia intracellularis* by Cuttino & McCabe in 1949 [34] and as a mycobacterium by Runyon in 1965 [10] (Table 1). In general, *M. intracellulare* is known to be more prevalent in clinical and environmental samples than *M. avium*, having a broader host range, and is responsible for disseminated diseases associated with MAC in patients with HIV [32,35].

In particular, *M. avium* subspecies pollute the environment following excretion from infected animals, which may be a source of infection, but no such epidemiological evidence has been provided in *M. intracellulare* [6]. Ten other recently reported species/subspecies that are closely related to *M. intracellulare* have also been linked to human disease, although there is insufficient data to explain the epidemiology, disease type, and host specificity, as well as a lack of information on the whole genome [6] (Table 1). Therefore, in this review, we would like to explain the genetics among *M. avium* subspecies that are genetically similar but show varying phenotypes.

### 2.2. Difference in Evolutionary Modes among MAC Subpsecies

The genetic markers underlying host specificity, variations in physiological characteristics, and disease types are not yet clear in *M. avium* subspecies [36]. Generally single genes such as *hsp65* and 16srRNA or insertion sequences such as IS*901* and IS*1245* were used for genetic identification of MAC [37,38,39,40,41]. For example, variable number tandem repeat (VNTR) is a genotyping method using serial numbers that represents repeat units numerically in gene loci with tandem repeats [42]. The identification of VNTR loci in MAC and genotyping techniques using these loci have been developed, and the epidemiological and clinical implications of genotypes have been explained using VNTR [6]. According to Radomski et al. (2010), in the minimal spanning tree using the VNTR profile, MAA, MAP, and MAS represent independently linked genotypes which have evolved independently from common ancestors [6,22].

Of note, many researchers have elucidated genetic information by performing whole genome sequencing of MAC organisms, and have thus investigated the relevance of different genetic signatures [37,43]. In comparative genomic analysis for nucleotides among MAC subspecies, the genetic similarity was high, with more than 97% nucleotide similarity between MAA and MAP [43]. As a result of phylogenetic analysis using the core genome, which consists of genes present in all strains, it was estimated that each of the MAC subspecies evolved from a common ancestor through its own evolutionary pathway [43]. In addition, the MAP strain showed the lowest sequence diversity, whereas the MAH strain had the highest sequence diversity, and it was therefore believed that gene acquisition and deletion through horizontal gene transfer occurred at a high rate in the evolutionary process in MAH [43]. Plasmid DNA was present in the MAH and MAA strains, but not the MAP strain, thus showing that MAH is an open pangenomic species [44,45].

The average genome size of MAH is large, and phylogenetic analysis has shown that MAP, MAA, and MAS were independently associated with MAH, but it undetermined which of the subspecies is closest to the ancestor [22,45]. In conclusion, while the MAP genome is very stable and lacks rearrangement due to small genetic variations, MAH could acquire new genes rapidly, contain plasmids, and have a larger gene repertoire than MAP [13,45]. This is the mechanism of adaptation to the niche of MAC subspecies and is considered to be most likely due to genetic evolution under selective or environmental pressure [13,45].

### 2.3. Genetic Differences in Host Specificity and Physiological Characteristics of MAC

Analysis of the phylogenetic relationship between the *M. avium* subspecies based on the single nucleotide variants, resulted in the division of this subspecies into approximately three distinct clusters, which were roughly identified as MAH, MAP, and MAA and MAS strains of avian origin, suggesting that this could be genetically explained in determining the host specificity of MAC [46]. In addition, Uchiya et al. (2015) characterized the pMAH135 plasmid derived from a MAH strain isolated from an HIV-negative patient with pulmonary MAC disease [47]. A number of virulence genes, such as mycobactin biosynthetic protein, a type VII secretion system-related protein involved in the pathogenicity of mycobacteria, and a domain presumed to be a multiple drug efflux transporter, were included in this plasmid [47]. As a result of southern hybridization of these genes, the genes were found to be present in human isolates, but rarely in pig isolates, suggesting that plasmids may affect pathogenicity and host specificity [47].

In addition, many studies have attempted to explain the main differences among the MAC subspecies through the PE and PPE family proteins unique to mycobacteria [37]. PE and PPE, which occupy 10% of the TB genome, are named after the motifs Pro-Glu and Pro-Pro-Glu found in domains conserved near the N-terminus of the protein, which are rich in GCs and are therefore thought to be the main cause of variability within the MAC species [36,37]. Most of the PE and PPE sequences showed only 82.9% and 79.7% identity to the PE and PPE sequences of *M. intracellulare*, but were conserved with mean nucleotide sequence identity of 99.1% and 98.1%, respectively, among the MAC subspecies [37]. In addition, although the nucleotide sequence was conservative, it could be confirmed that a unique amino acid sequence was generated due to amino acid substitutions and frame shifts, and it is believed that such genetic variation could make result in differences between subspecies [37]. In addition, Timm et al. (2015) reported a difference in PE/PPE genes according to host origin by conducting a genome analysis with MAP human isolates and other mycobacterial pathogens, and although the function is not yet known, the genetic possibility of the host specificity of the PE/PPE genes was investigated [36].

On the other hand, the growth rate of MAC, which is an SGM (slow growing mycobacteria), varies greatly among subspecies [13]. The mammalian cell entry (mce) gene, which is present in many bacterial species, but is known to exist only as an operon in mycobacteria, has been reported in studies on the invasion and survival of TB in host non-phagocytic cells [36,48]. In particular, the mce3 mutation of TB grew more slowly than that of the wild type, and the mce3R deletion in the mce3 operon was found in MAP among MAC subspecies [36,48]. Therefore, it was thought that this could be a possible explanation for the long incubation time of MAP. [36]. Bannatine et al. (2003) genetically compared the oriC region related to chromosome replication, which can directly affect the growth rate, but it was not possible to explain the difference in phenotype for MAC subspecies as this region was found to be highly conserved [43]. In addition, a number of potential explanations have been discussed to elucidate the reasons for the complete phenotypic differences, even with the high nucleotide similarity between MAC subspecies, including the presence of transcription-translation rates, insertion sequences, and genome rearrangements [43]. Although there was no significant difference in transcription and translation rates compared to MAP and *M. smegmatis*, since MAC has an insertion element (IS*900*, IS*1311*) at its own locus, insertion mutations have been shown to exert a distinct effect on growth differences or other phenotypes [43,49]. In addition, at least one large-scale genomic rearrangement can be identified between MAC subspecies. As mentioned above, a unique amino acid sequence may be generated due to amino acid substitution and frame shift, which is expected to have a great influence on the phenotype [37,43].

Mycobactin dependence in vitro is also a major phenotypic difference between MAP and other MAC subspecies [36]. Mycobactin is a siderophore that transports iron, especially in environments where free iron is restricted, such as inside host cells [50]. MAC subspecies are exposed to a host environment that is deficient in nutrients, including iron; among these subspecies, MAP has demonstrated mycobactin-dependent growth in vitro [36]. The dependence on mycobactin has been studied by comparing mycobactin gene clusters, with an emphasis on differences in the length of genes in the cluster as well as the gaps between genes [36,51]. Whole-genome sequencing of MAP K10, which exhibits mycobactin dependence, revealed that the homolog of *mbtA*, which encodes the first enzyme acting on salicylic acid in the mycobactin biosynthetic pathway, is truncated, potentially inactivating mycobactin production [51,52]. *MbtF* encoded a protein that was found to be shorter in mycobactin-dependent MAP than in mycobactin-independent MAP, MAH, MAA, and MTB [36].

Interestingly, mycobactin dependence affects antimicrobial susceptibility with respect to iron metabolism in MAC [53]. Under iron-restricted conditions, the resistance of MAH to macrolides, aminoglycosides, and levofloxacin-related fluoroquinolones increased whereas the susceptibility to isoniazid, ethambutol, and D-cycloserine, which target cell wall synthesis, increased [53]. Metabolic changes could notably alter the antimicrobial susceptibility profile, suggesting that the difference in metabolism may play a role in the antimicrobial susceptibility [53].

### 2.4. Genotypic and Genetic Explanation of MAC (Sub)Species in Human Disease

There have been several studies demonstrating the genetic diversity and geographic differences in MAH using VNTR analysis [6,46]. In particular, Uchiya et al. (2017) suggested that geographic differences in MAH genotypes are the reason for the frequent occurrence of lung MAH-PD in Japan [46]. In addition, the VNTR results of isolates from MAH-PD patients confirmed that isolates from patients with advanced disease were grouped into specific clusters, suggesting that this may indicate disease type and severity according to genotype [46]. In particular, in a cluster consisting of many MAH isolates from patients with advanced disease, eight isolates contained the pMAH135 plasmid and specific genetic factors including the *mce* family gene and the *mmpL* gene [46]. Therefore, it was believed that the genes encoding the virulence genes of these strains were acquired through splicing and horizontal transfer, resulting in disease progression [46]. The genes in pMAH135, were detected more frequently in the isolates of MAC-PD patients than HIV-positive patients [46,47]. Therefore, it could be inferred that the genes in pMAH135 may affect the pathologic expression of MAC disease [46]. In addition, Uchiya et al. (2013) and Jeffrey et al. (2017) attempted to explain the genetic differences in virulence genes for the different MAH disease types, including PD and disseminated disease, using isolates of MAH-PD patients and HIV-positive patients [44,54]. First of all, the MAH which cause PD invade through the respiratory mucosa and survive in alveolar macrophages, while strains which cause disseminated disease are usually acquired via the gastrointestinal route [44,55]. Bacteria that pass through the gastrointestinal tract and invade the lamina propria are phagocytized by phagocytes and then spread into the blood through the lymphatic vessels and are absorbed by the spleen and liver [44,55]. Therefore, cell invasion of MAH is an important process in establishing disseminated disease, and the genes encoding Mce family proteins, which are related to cell invasion, showed low homology among strains according to disease types [44]. In particular, it was characteristic that the MAH104 strain of the disseminated disease type contained more genes encoding the strain-specific Mce protein than the strain TH135 of PD type [44]. Meanwhile, the ability to replicate in phagocytes associated with the establishment of chronic PD is important, and there was an additional catalase gene (MAH_4495) exclusive to the TH135 strain [54]. In addition, the strain TH135 possessed more *mmp* genes than the strain 104; these genes are essential for lipid transport through membranes, secretion of host-regulated lipids, and maintenance of the surface of bacterial cells, suggesting a difference in the composition of cell wall lipids between these two strains [44]. Therefore, although the function of specific genes was not revealed, genetic differences were shown in strains according to disease types, and it was speculated that these may show the association of bacterial factors related to pathological signs of MAH disease [44,54]. However, as explained in the evolutionary aspect of MAC above, MAH has a larger gene repertoire than other subspecies, resulting in rapid acquisition and removal of new genes by adapting to various environmental pressures as it can survive in various environments. Thus, a single MAH strain cannot be representative, and the genetic characteristics for a phenotype representative of a single MAH strain cannot be easily concluded. Therefore, the genetic explanation of the strategy for surviving and overcoming the host cell environment in the early stage of infection of MAH will be described in detail in the next section, based on the clear identification of genes.

## 3. Virulence Gene-Associated Adaptation Strategies of MAC during Pathogenesis

*M. avium* subspecies can survive in various hosts, ranging from environmental regions to amoebas, animals, and humans. In particular, among MAC, MAH shows a wide range of genetic mutations, which enable its survival in a wide spectrum of environments and hosts [6,7]. In other words, the diverse survival mechanisms adopted by MAH to survive in various environments are thought to lead to a wide range of genetic evolution. Therefore, it is necessary to understand the interaction between the human host and MAH and to understand the genetics underlying the mechanisms by which MAH can escape the host defense mechanisms [9].

MAC can pass through the mucosal barrier and infect macrophages to replicate within those cells [56]. In response to MAH infection, macrophages show innate defense mechanisms such as production of superoxide anions, nitric oxide, and antimicrobial peptides and induction of autophagy [56]. However, MAC can survive from such defense mechanisms by inhibiting the acidification of the phagosome, fusion of phagosomes and lysosomes, and influx of toxic compounds into the phagosome [57,58]. As a last resort, MAH-infected macrophages undergo apoptosis to kill the pathogens [56]. However, macrophage-induced apoptosis can be avoided by MAH in many cases, and MAH can instead promote apoptosis in order to disseminate the infection to other cells [56,59,60]. In this section, the strategies adopted by MAC for host invasion, manipulation of the immune response, and survival in host cells against the host’s defense mechanisms will be described, using relevant genes (Table 2).

### 3.1. Mucosal Epithelial Cell Adhesion and Invasion

Similar to various other pathogens, the formation of microaggregates and biofilms can be the first key factor in the infection and persistence of infection by MAH [61]. Several studies have reported that MAH can infect drinking water sources around patients [6,62,63]. MAH are also able to inhabit and form robust and complex biofilms in various sources of drinking water, including distribution pipes, bathtub inlets, faucets, showers, swimming pools, and hot tubs, and studies have demonstrated that MAH could also form biofilms in urban PVC pipes [62,63]. Moreover, MAH can mainly infiltrate the respiratory or intestinal mucosa of the host to cause infection [44,55]. It can bind to and cross the mucous membrane, while simultaneously avoiding the host defense, and this leads to the interaction of MAH with the host [61] Therefore, the following section will explain the strategies adopted by MAH to colonize the airways through the formation of microaggregates and biofilms.

MAH are characterized by surface-expressed proteins that interact with the host proteins to promote adhesion to host epithelial cell membranes [61,64]. MAC infection is prevalent after secondary lung damage and in patients with chronic obstructive diseases, such as cystic fibrosis and nodular bronchiectasis, and it was demonstrated in patient tissues that *M. avium* can bind to damaged and non-ciliated epithelium [65]. Furthermore, fibronectin, which is commonly observed in the plasma and extracellular matrix (ECM), plays a role in binding to the damaged epithelium through the exposed ECM [65], and has been shown to bind almost exclusively to fibronectin adhesion protein (FAP), Ag 85 complex, and MPA51 of MAH [64,66]. In the studies by Yamazaki et al. (2006), when MAH was exposed to human respiratory epithelial cells for 24 h, microaggregates composed of 3–20 bacteria were formed, and these were shown to invade respiratory epithelial cells with a higher efficiency than planktonic bacteria [67,68]. Barak et al. assessed a series of processes for the formation and invasion of microaggregates and related genes in two studies [61,69]. In the formation of microaggregates, microaggregate binding protein 1 (MBP-1, MAV_3013) and microaggregate invasion protein-1 (MIP-1, MAV_0831) were highly expressed, and were found to be related to the formation of microaggregates and an increase in the ability to bind and invade respiratory epithelial cells [61,69]. It was observed that MBP-1 interacted with MAV_4504, an ABC transporter ATP binding protein, and during formation of microaggregates, was thought to migrate or fix MBP-1 to the bacterial surface [61]. In particular, MBP-1 bound to vimentin, a host cytoskeletal protein, and microaggregates could not bind to host cells after treatment with anti-vimentin antibodies [61]. This suggested that MBP-1 is a major route of attachment by inducing the formation of MAH microaggregates and polymerization and phosphorylation of vimentin in the host [61]. Moreover, as described above, MAC infection is prevalent after secondary lung injury in chronic obstructive diseases, and it was thought that overexpression of vimentin in damaged lungs was highly related to MAH adhesion and infection [58,70]. Once MAH attached to host epithelial cells via MBP-1, using vimentin as an adhesion receptor, the surrounding planktonic bacteria were rapidly recruited, and microaggregates were formed within 24 h. Subsequently, MAH microaggregates were able to efficiently invade the host epithelium by interacting with the host protein filamin A through MIP-1 and further manipulate the host cytoskeleton [69] (Table 2, Figure 1). 

Members of the MAH family have mechanisms to invade the epithelial cells by regulating the signaling pathways of the host cells. FadD2 is one of the genes that can control skeletal rearrangement of the host cells; it encodes the fatty acyl coenzyme A synthase, which is involved in the breakdown of fatty acids [71,72]. In human cells, the small G-protein, Cdc42, indirectly activates N-WASp via phosphorylation, subsequently causing N-WASp to bind and activates the Arp2/3 complex in order to induce actin polymerization. It was observed that fadD2 could regulate the activation of the host cell Cdc42 signaling pathway [73]. Moreover, when the fadD2 gene was exposed to HEp-2 cells, it indirectly regulated several genes, including transcriptional regulators, membrane proteins, and secreted proteins, and among them, CipA interacted by activating Cdc42 [74]. In summary, it was assumed that the invasion of MAH into epithelial cells is partially regulated by fadD2 and other downstream transcriptional regulators, and that the mechanism of invasion involves the activation of actin polymerization through the interaction between the bacterial surface and host cell membrane for effective invasion into mucosal epithelial cells [74] (Table 2, Figure 1).

Formation of biofilms can be a highly important survival strategy for MAH that inhabit various environmental and host conditions. In studies on biofilms, MAH A5 strain, which produces many biofilms, is the main model [67]. Aggregates of MAH A5 were formed 2 h after infection, and these cells produced more biofilms when cultured on PVC surface than on 7H9 medium [67]. It was observed that during the formation of biofilms, microorganisms hardly came into contact with clean surfaces and generally absorbed molecules such as water and proteins from the environment to form colonies on the modified surface [67]. Krzywinska and Schorey (2003) explained the genomic differences, especially in the GPL gene cluster, between MAH 104 and MAH A5 [75]. In particular, Yamazaki et al. (2006) screened genes that were regulated during the formation of biofilms and showed that genes related to GDP-mannose and GPL biosynthesis encoded proteins which participate in fatty acid biosynthesis, suggesting that the extracellular bacterial surface may be important for formation of biofilms [67] (Table 2, Figure 1).

In particular, the ability of MAH to form biofilms in the host tissue was involved in infecting bronchial epithelial cells. This function was not eliminated by the innate immune system in mice and was associated with early establishment in the respiratory tract [67,76]. Unlike planktonic MAH infection, macrophages produced high levels of TNF, superoxide, and nitric oxide after exposure to MAH biofilms [77], and this biofilm partially decreased macrophage function and induced apoptosis through TNF-based hyperstimulation to prevent removal of biofilms by innate immune cells, suggesting that biofilm-related infection could persist [76]. The structure of biofilms also prevents the optimal penetration of antibiotics and interferes with the mechanism of cell killing by drugs [78,79]. Resistance to clarithromycin, azithromycin, and moxifloxacin, which are commonly used to treat MAH biofilm infection, have been observed [67,80], and Rojony et al. (2019) showed that the LrpB lipoprotein, which is essential for *M. tuberculosis* virulence and survival in vivo, was highly expressed in MAH biofilms in the clarithromycin and amikacin-treated experimental groups [78]. In addition, LrpB overexpressing clones were more resistant to anti-bacterial agents than the wild type strain [78]. Therefore, it was suggested that not only the presence of antibiotics, but also changes in MAH caused by environmental conditions can transport factors related to biofilm formation and toxicity and contribute to bacterial resistance in new environments (Table 2).

**Table 2 ijms-22-03011-t002:** Virulence-associated genes of *Mycobacterium*
*avium* subsp. *hominissuis* during the pathogenesis.

Host Defense	Gene	Description	Strategy of MAH	References
Action	Description	Strain
Strategy 1. Mucosal epithelial cell adhesion and invasion
***Mucus layer/mucosal epithelial cell***	ag85mpa51	- antigen 85- *M. avium* MPB51	attachment	▪ Binding to fibronectin▪ Epithelial cell adhesion using host protein	MAH ATCC 15769	[64,66]
MAV_3013(MBP-1)	- microaggregate binding protein 1	invasion/ attachment	▪ Binding to vimentin▪ Invasion of mucus layer▪ Epithelial cell adhesion using host protein▪ Formation of microaggregate	MAH 104	[61,69]
MAV_4504	- ABC transporter, ATP-binding protein coordinates	transport	▪ Translocation of MBP-1 to the bacterial surface	MAH 104	[69]
MAV_1799	- hypothetical protein	aggregation	▪ Rapid recruitment of planktonic bacteria▪ Formation of microaggregate	MAH 104	[69]
MAV_0831(MIP-1)	- microaggregate Invasion Protein-1	invasion/attachment	▪ Binding to flaminA▪ Invasion of epithelial cell	MAH 104	[69]
*fadD2* *cipA*	- fatty acyl coenzyme A synthase- a domain similar to the PXXP motif of the human piccolo protein	invasion	▪ Activation of Cdc42 signaling pathway▪ Rearrangement cytoskeleton	MAH 109	[72]
*accA2,* *sucA,* *pstB*	- acetyl/propionyl-CoA carboxylase (subunit)- 2-Oxoglutarate dehydrogenase- protein synthetase	attachment	▪ Association with GPL biosynthesis▪ Formation of biofilm	MAH A5	[67]
*guaB2* *gtf* *pmmB*	- IMP dehydrogenase- glycosyltransferase- Mannose-1-phosphatase	attachment	▪ Association with GDP-mannose and GPL biosynthesis▪ Formation of biofilm	MAH A5	[67]
*-* *Pcd*	- hypothetical membrane protein- piperideine-6-carboxylic acid dehydrogenase	attachment	▪ Association with biosynthesis of aminoadipic acid▪ Formation of biofilm	MAH A5	[67]
LprB	- leucine-responsive regulatory protein B	cell wall	▪ Bacterial cell surface protein▪ Component of biofilm matrix▪ Association with antibiotic resistance	MAH 104	[78]
MAVA5_03380MAVA5_10375	- FtsK/ SpoIIIE-like DNA translocation protein	transport	▪ Export of eDNA▪ Component of biofilm matrix▪ Association with antibiotic resistance	MAH A5	[81]
MAVA5_19945MAVA5_22765	- carbonic anhydrase	-*	▪ Export of eDNA in reseponse to bicarbonate▪ Component of biofilm matrix	MAH A5	[82]
Strategy 2. Resistance to the phagocytic environment in the immune cells
***Oxidative stress/*** ***phagosome acidification***	MAV_2043(Cu-Zn SOD)	- Cu-Zn superoxide dismutase	catalysis	▪ Protection from oxidative stress	MAH 104	[83]
MAV_2839(ahpC)	- Alkyl hydroperoxide reductase	catalysis	▪ Protection from oxidative stress	MAH 109	[84]
MAV_4682(aceA)	- isocitrate lyase	metabolism	▪ A key enzyme in glyoxylate shunt▪ Use of fatty acids and acetates as basic carbon resources under carbon restricted conditions	MAH 109	[84]
MAV_2450(pks12)	- Polyketide synthase 12	cell wall	▪ Association with susceptibility to oxidative products▪ Inhibition of phagosome acidification	MAH 104	[85]
MAV_4292	- Hypothetical protein	-*	▪ Association with susceptibility to oxidative products (es, nitric oxide)▪ Inhibition of phagosome acidification	MAH 104	[85]
MAV_4012	- Conserved hypothetical protein	-*	▪ Association with susceptibility to oxidative products▪ Inhibition of phagosome acidification	MAH 104	[85]
MAV_4264	- Hypothetical protein, homology with bacterial regulatory protein TetR domain	-*	▪ Regulation of the genes that participate in the inhibition of phagosome acidification (ex, MAV_2450, MAV_4292, MAV_4012)▪ Inhibition of phagosome acidification	MAH 104	[85]
MAV_4644	- putative pore-forming protein that has ADP-ribosyltransferase (ADPRT) activity	interfering with host peptide	▪ Binding to cathepsin Z▪ Interfering with CTZ action that induces bacterial death in the presence of NO	MAH 104	[86]
***Phagosome-lysosome fusion***	MAV_1356	- calmodulin-like protein	hijacking host protein	▪ Hijacking to Annexin A1 and S100-A8▪ Regulating the phagocytic membrane▪ Blocking phagosome-lysosome fusion	MAH 104	[87]
MAV_2928	- PPE25-MAV	secretion	▪ Responsible for the Esx-5 region of the Type VII secretion system▪ Export system for adjacent ESAT family gene, MAV_2921▪ Interfering with endosome maturation	MAH 109	[88,89]
MAV_2941	- Hypothetical protein, a small protein only present in *M. avium*	hijacking host protein	▪ Hijacking host trafficking proteins (ex, AP3B1, STX8 and ARCN1)▪ Interfering with endosome maturation	MAH 104	[90]
*oppA*	- Oligopeptide transporter	transport	▪ Active transport of oligopeptides and small protein (ex, MAV_2941)	MAH 104	[91]
Strategy 3. Resistance to antimicrobial peptide
***Secretion of antimicrobial peptide***	*lysX*	- lysyl-transferase-lysyl-tRNA synthetase	lysinylation	▪ Association with GPL expression▪ Resistance to human beta defensin-1	MAH 104	[92,93]
MAV_0216	- Cutinase superfamily protein	-*	▪ Resistance to antimicrobial peptide (polymyxin B)	MAH 104	[94]
MAV_3616	- Long-chain specific acyl-CoA dehydrogenase	-*	▪ Resistance to antimicrobial peptide (polymyxin B)	MAH 104	[94]
MAV_2450	- Erythronolide synthase (polyketide synthase), modules 3 and 4	-*	▪ Resistance to antimicrobial peptide (polymyxin B)	MAH 104	[94]
Strategy 4. Induction of immune cell death and spreading tactics
***Cell death***	MAV_2052	- putative cysteine synthase A protein	induction of cell death	▪ Induction of cell death through TLR4-dependent ROS production and JNK pathway	MAH 104	[95]
MAV_2054(MMP-1)	- 35-kDa major membrane protein 1	induction of cell death	▪ Induction of cell death via ROS production and the mitochondrial pathway	MAH 104	[96,97]
MAVA5_06970	- a secreted protein	induction of cell death	▪ Hijacking OPN to hinder the operation of OPN▪ Induction of cell death and limitation of the activation of the type I immunity pathway▪ Enhancement of bystander macrophage	MAH A5	[59]

* This gene is related to the MAH survival strategy for the host defense, but its mechanism has not been confirmed yet.

Several components of the unique mycobacterial biofilm matrix, including extracellular DNA (eDNA), free mycolic acid [98], glycopeptideolipid [99], and other lipid-containing molecules [100] were identified [67,82]. DNase I treatment reduced colonies of biofilm in vitro, was effective in removing established biofilms, and reduced MAH resistance to antimicrobial agents [81,101]. As such, the release of eDNA in biofilms, which greatly contributes to colonization, persistence, and drug resistance of biofilms, is considered an important survival strategy for MAH [81]. RAPD analysis showed that eDNA of MAH A5 biofilms had similar genomic origins as genomic DNA, and it is still unclear whether eDNA are products of cell lysis or actively secreted [81]. However, it was recently reported that eDNA was produced and secreted in MTB to escape perforated phagocytosis and stimulate the cytoplasmic surveillance pathway in order to promote infection [76]. Rose et al. (2015) quantified eDNA using MAH A5 cells and observed almost no differences in CFU when eDNA was increased dramatically, while lysed cells were rarely observed in micrographs of initial biofilm [81]. Thus, it was thought that eDNA was produced and secreted rather than being a product of cell lysis [81]. In addition, Rose & Bermudez (2016) demonstrated that respiratory epithelial cells secrete bicarbonate from the airway surface fluid, which can be detected by MAH to promote eDNA production and secretion [82]. These findings suggested that MAH could export eDNA under certain conditions, and that bicarbonate could act as a derivative in eDNA release [82]. Furthermore, MAH DNA induced IL-12 and TNF-in a TLR9-dependant manner and that MAH-infected TLR9 knock out (KO) mice exerted an increased bacterial burden in the organs, which confirmed that MAH DNA contributes to the immune response through TLR9 [102]. Therefore, it was thought that the inherently high TNF-α production induced by the biofilm produced by MAH A5 could be partially explained by the immune response of MAH eDNA by TLR9 [76,81]. 

### 3.2. Survival Strategy in Phagocytic Immune Cells: Survival Strategy within Phagosomal Environment in the Early Stage of MAH Infection

MAC is an intracellular pathogen that targets macrophages to avoid the host’s innate immunity and survive long-term in the host. Macrophages generally undergo a series of programmed events including induction of reactive oxygen species and nitrogen intermediates, gradual acidification of phagosomes, phagosome-lysosome fusion, and antigen processing upon absorption of bacteria, resulting in most of the bacteria being eliminated through these processes [103,104]. However, pathogenic mycobacteria can survive using the macrophages as an important haven after phagocytosis thanks to their survival strategies. Therefore, we will now discuss the strategies of MAH for survival after phagocytosis by macrophages.

Above all, proteins on the extracellular surface of MAH are likely to play an important role in overcoming the host immune response, and surviving, and replicating in host macrophages of the early stages of infection [84]. This is because the extracellular proteins are generally known to have roles in adhesion, motility, molecular transport and conjugation [83]. In several studies that have simulated early infections, extracellular surface proteins, among many survival-related factors of MAH, were characterized. It was reported that neutrophils, in addition to macrophages, were involved in effective innate responses to MAH in only the early stages of infection [83,105,106]. Neutrophils and macrophages produce and release large amounts of superoxide anions, and Cu-Zn superoxide dismutase (MAV_2043) was characterized as a strategy by MAH to avoid such defense mechanisms [83]. Interestingly, MAV_2043 (Cu-Zn SOD) is a surface protein that is expressed even before contact with cells, and MAV_2043 defiance greatly reduced the survival of MAH in neutrophils, while it only slightly decreased survival in macrophages [83,107]. This suggested that MAH avoids neutrophils and preferentially infects macrophages in the early stages of infection [83]. Furthermore, McNamara et al. (2012) reported that alkyl hydroperoxide reductase (ahpC, MAV_2839) and isocitrate lyase (aceA, MAV_4682) were uniquely expressed after MAH were exposed to macrophages [84]. In particular, ahpC encodes a protein that catalyzes peroxide reduction, and exhibited resistance to oxidative stress, suggesting that it is an essential factor for intracellular survival [84,108]. This was also highly upregulated after phagocytosis of TB by THP-1 macrophages [109]. Isocitrate lyase is a key enzyme in glyoxylate shunt [105], and under carbon restricted conditions, mycobacteria can directly use fatty acids and acetates as basic carbon resources through glyoxylate shunt [84]. This may be a key process in the macrophage environment with limited nutrition, and it has been reported as an essential persistence factor for Mycobacterium infection in both macrophages and animal models [84,106]. In addition, modD, which is thought to play an important role in bacterial adhesion to the extracellular matrix, was present when the cells were cultured in medium and absent after exposure to macrophages [84]. Thus, it was observed that MAH actively tries to survive in toxic intracellular environments by changing the surface proteins after exposure to macrophages (Table 2, Figure 2). 

MAH appears to be actively involved in building an intracellular environment for replication and survival within the phagosome. The macrophage phagosome with phagocytosed bacteria is established as an early endosome through acidification, and this leads to the characteristic expression of the transferrin receptors and Rab 5 [88]. Early endosomes undergo maturation to acquire late markers such as LAMP-1, Cathepsin D, Rab 7, and calmodulin, and this leads to phagosome-lysosome fusion [88]. In particular, acidification of endosomes/phagosomes is an early anti-bacterial mechanism in macrophages. Li et al. (2010) identified genes that inhibited acidification in phagosomes that were related to resistance to oxidative stress in macrophages, and all of the KO strains for these genes were attenuated in the early stages of infection in mice [85]. MAV_4264, which has high homology with the bacterial regulatory protein TetR domain, was shown to regulate the inhibition of acidification by other genes (MAV_2450, MAV_4292, and MAV_4012) [85]. Thus, it was observed that several hypothetical proteins, in addition SOD or correlated proteins, were related to resistance to superoxide anion and reactive nitrogen intermediates [85] (Table 2).

Furthermore, it was demonstrated that inhibition of Ca++ signaling in TB is related to the ability to interfere with phagocytic fusion, suggesting that MAH may have a similar strategy [110]. Chinison et al. (2016) observed that MAH secreted several proteins under conditions that mimicked the metal ion concentration and pH of the phagosome, and among secreted proteins, MAV_1356, which interacts with the host proteins Annexin A1 and S100-A8, was identified [87]. Annexin A1 is a factor that binds to the phagosome in the presence of calcium and promotes the interaction of F-actin in the phagocytic membrane, while S100-A8 is a macrophage protein that binds to calcium [87,111]. Interestingly, MAV_1356 was a hypothetical protein homologous to Rv1211 calmodulin-like protein of *M. tuberculosis*, and it was shown that Rv1211 blocked phagosome-lysosome fusion by inhibiting the entry of cytoplasmic Ca++ inside vacuoles and interfering with Ca++ calmodulin-mediated downstream signaling of macrophages [87,110]. Therefore, it was thought that MAH MAV_1356 interacts with Annexin A1 and S100-A8 in the phagosome to play various roles in the intracellular environment, such as regulating the cytoskeleton and blocking phagosome-lysosome fusion [87] (Table 2, Figure 2).

Li et al. (2005) assessed the function of MAV_2928 among the Mycobacterium-specific proteins of the MAH PE/PEE family [88]. MAV_2928, which is 52% homologous to Rv1787 of TB, is located within the ESX-5 region of the type VII secretion system [88,112], and it was observed that the conserved N-terminus domain of MAV_2928 was located on the bacterial surface and interacted with the adjacent ESAT family gene, MAV-2921, through the C-terminus domain [89]. MTB exports important virulence factors such as CFP-10 and ESAT-6 through the ESX-1 system [113]. Because ESX-1 is absent in MAH, it was observed that ESX-5, which is encoded by the MAV_2928 gene, compensates for the EST family protein to export these proteins [89]. Most of all, MAV_2928 inhibited the expression of EEA-1, Rab-5, and Rab7, thereby interfering with the maturation of endosomes [88]. In addition, while MAV_2928 was not observed in the culture medium, it was expressed after phagocytosis on macrophages, and was particularly upregulated under nutrient limited conditions [89]. This suggests that the mechanism via MAV_2928 is to establish a suitable environment for survival in the macrophage phagosome [88,89]. Another gene, MAV_4644, in studies by Lewis et al. (2019), was shown to exhibit resistance to nitric oxide and was secreted into the VII secretion system to bind to the outer membrane of MAH [86]. MAV_4644 is located in the same operon as MAV_4643 and MAV_4642, which are thought to be homologues of Esat-6 and CFP-10 of TB, and it was thought to be a virulence-related effector protein, secreted by the VII secretion system in a similar manner to MAV_4643 and 4642 [86]. In particular, the MAV_4644 gene was expected to be a putative pore-forming protein that has ADP-ribosyl transferase (ADPRT) activity, and it was observed to interact with the host lysosomal peptidase cathepsin Z (CTSZ) [86]. CTSZ is an important regulator of inflammation and apoptosis, and although it did not have direct killing effects on MAH, it contributed to apoptosis of bacteria in the presence of nitric oxide (NO) [86,114]. Therefore, it was believed that the secreted gene product of the MAV_4644 operon plays a role in protecting MAH from CTSZ by translocating to the mycobacterial membrane in the early stages of infection [86] (Table 2, Figure 2).

### 3.3. Hijacking Tactics of Host Trafficking Protein

Danelishvili and Bermudez (2015) showed that MAH has a unique strategy to inhibit maturation of phagosomes by hijacking host trafficking proteins in macrophages [90]. In TB, MAV_2941 is absent, while in MAP, the open reading frame (ORF) is cleaved such that the protein is not produced. In contrast, in MAH, a small amount of MAV_2941 is expressed [90]. Interestingly, MAV_2941 is structurally homologous to human phosphatidylinositol 3-kinase (PI3K) and interacted with syntaxin-8 (STX8), subunit beta-1 of adaptor-related protein complex 3 (AP3B1) and Archain 1 (ARCN1), all of which are vesicle trafficking proteins [90]. AP3B1 is a protein that transports trafficking cargo proteins including lysosome-related membrane protein 1 (LMAP-1) to the phagosome and lysosome-related organelles, while STX8 is a main component of the SNARE complex that is involved in intracellular trafficking and mediating membrane fusion, and is also known to play a role in regulating fusion of late endosomes [90]. Additionally, ARCN1 is a cytoplasmic vesicle transport protein that participates in protein transport from the endosomal reticulum (ER) to the trans-Golgi network [90].

MAV_2941, with mutations in the PI3K homology region did not interact with the trafficking proteins, and transfection of macrophages with MAV_2941 significantly reduced the colocalization of MAH phagosomes and LAMP-1 [90]. These results proved that MAV_2941 competes with PI3K to hijack trafficking proteins and interfere with transport in order to hinder maturation of phagosomes [90]. Furthermore, MAV_2941 was secreted into the macrophage cytoplasm by OppA, which belongs to the ATP-binding cassette (ABC) transporter family, and OppA was highly expressed in tissues of MAH-infected mice [91]. Therefore, it was expected that MAH would actively utilize the strategy of hijacking trafficking proteins of host cells via MAV_2941 for survival (Table 2, Figure 2).

### 3.4. Release of OMV for Sustained Infection

Recently, outer membrane vesicle (OMV) in MAH was analyzed as a strategy for release of effector proteins in order to establish colonization and survival niche in vivo [115]. In an in vitro model of mycobacterial phagosomes that mimics phagosomal pH and metal ion content, MAH secreted vesicles rich in several known virulence factors and enzymes related to metabolism of lipids, fatty acids and DNA [115]. Characteristically, anti-oxidant enzymes, including trehalose (TDM) synthase, SOD, catalase, catalase peroxidase, and ahpC that synthesize TDM, PE/PPE family proteins, and dsDNA were also included in the vesicles [115]. TDM is a main pathogenic factor of mycobacteria related to immune evasion and tissue damage through association with host lipids [116], and PE/PPE family proteins are abundant in mycobacteria and are pathogenic factors in many processes, including macrophage uptake, inhibition of phagocytic maturation and acidification, secretion of cytokines, and stimulation of host cell apoptosis and death [115]. In an in vitro model of mycobacterial phagosome, various antioxidant enzymes contained in OMV recruited during the first 24 h after infection were secreted, suggesting that MAH exports effector proteins through OMV as an early reaction to macrophage infection [115]. Conclusively, it was suggested that MAH vesiculation was triggered under conditions of phagosomal environment and used OMV as a delivery mediator of several MAH virulence-related products within phagocytic cells (Figure 2).

### 3.5. Genes Related to the Tolerance to Attacks of Host Antimicrobial Peptide

The host releases antimicrobial peptide molecules into cells in the mucosal surface, macrophages, and neutrophils to eliminate pathogens, and it was already demonstrated that human defensins have in vitro bactericidal and bacteriostatic activity against MAH and MTB [94,117,118]. Significant levels of β-defensins were observed in the bronchoalveolar lavage fluid of MAH-infected patients, and cathepsin D was localized in MAH-infected macrophages in an immature pro-enzyme form, suggesting a controlled protein environment in the endosomes [93,119]. Human cathelicidin (LL-37) was expressed in macrophages by stimulation of 1,25 vit D3, and this resulted in secretion of TNF-a, granulocyte-macrophage colony stimulating factor and antimicrobial peptides, inducing apoptosis of MAH [120].

In addition, Motamedi et al. (2014) screened mutants susceptible to high concentration of polymyxin B as a surrogate for host antimicrobial peptides in the MAH mutant library, and most of the identified genes were those related to cell wall synthesis and permeability, and most of them were also vulnerable to LL-37 [94]. Therefore, these results suggested that the envelope of MAH is the primary defense mechanism against host antimicrobial peptides [94]. In a study by Kirubakar et al. (2020), MAH mutants strains deficient in lysX, a lysyl-transferase-lysyl-tRNA synthase, showed higher levels of inflammatory cytokines (IL-1β, IL-12, TNFα, and IL-10) and a decreased viability against human β-defensins [93]. This suggested that the lysX gene was not only related to resistance to cationic antimicrobial agents, but also had immense effects on carbohydrate and lipid metabolism. LysX differentially affected the expression of genes involved in glycerophospholipid metabolism (MAV_1825, fbpB, fbpC) and GPL synthesis (MAV_4518, rmt4/mftb, fadE5, sap) [92,93]. TLC analysis showed the existence of changes in the structure of the GPL in a lysX mutant strain, thus showing that the lysX gene of MAH is related to GPL, which is a major pathogenic factor in MAC organisms, and plays a role in the regulation of major virulence of MAH [93,121]. Additionally, it was proven that most cell surface phospholipids in MTB are lisinylated and show resistance to antimicrobial peptides [94,122]. Therefore, it was suggested that the cell wall is involved in the primary defense against the host antimicrobial peptides and the resulting regulation of toxicity serves as a mycobacteria survival strategy (Table 2).

### 3.6. Induction of Immune Cell Death and Spreading Strategies

Macrophage apoptosis is generally thought to be a host defense mechanism activated to prevent the spread of mycobacterial infection [95]. In addition, it has been reported that 90% of MAC loses their viability upon apoptosis of host macrophage [123]. However, Pais et al. (2004) observed that there were only minor changes in the viability of MAC when macrophages were treated with staurosporin, an apoptosis inducing factor [124]. In addition, evidence that some bacteria escape from macrophages undergoing apoptosis or inhibit apoptosis in MAP infection has been presented [57]. In the case of MTB, several strategies have been reported to simultaneously induce and inhibit apoptosis in host cells [95,125]. As in MAC, apoptosis is partially effective at killing bacteria. However, it may not be the main mechanism by which the macrophages kill MAH, and may instead be an adaptive approach to manage apoptosis [57,126]. The exact role of macrophage apoptosis in MAC infection is unclear. Therefore, the following section describes factors of MAH that regulate apoptosis and the role of macrophage apoptosis in MAC infection.

First, MAC-infected macrophages are characterized by TNF- and Fas-mediated apoptosis [127], and these macrophages undergo apoptosis through caspase 8 activation via the ASK1/p38 MAPK signaling, caspase 9 activation [128] via mitochondrial death signaling, and the ER stress-mediated IRE1α-RIDD pathway [95,129]. In a study by Lee et al. (2016), it was observed that among MAH culture filtrate proteins, MAV2052 induced apoptosis through TLR4-dependent ROS production and ASK1/JNK pathway [95]. In the presence of anti-TNF antibodies, MAV2052-mediated apoptosis was not reduced, and mitochondrial translocation of BAX and release of cytochrome c from mitochondria were observed in macrophages treated with MAV2052, explaining that apoptosis is induced through a mitochondrial dependent pathway [95]. Furthermore, the MAV2054 protein also induced apoptosis through ROS production, the ASK1/JNK pathway, and the mitochondrial pathway [96]. In particular, the amino acids of MAV2054 were 92% and 97% homogenous to the matrix metalloproteinase 1 (MMP-1) of MAP and *M. leprae*, respectively, whereas not in other mycobacteria such as MTB and BCG [96]. In addition, since MAV2054 showed a strong immune reactivity in MAC-PD patients, it was suggested that it is involved in the regulation of cell death and pathogenesis inherent in MAC [97] (Table 2, Figure 2).

Second, it was suggested that the spread and transmission of MAH infection requires escape from macrophages and could possibly infect other macrophages [85,130]. Bermudez and colleagues have already shown that the action of MAH escaping from primary macrophages and entering secondary macrophages may decrease the number of bacteria. However, they also suggested that this phenotype is involved in the composition of dominant bacteria, which is related to the spread and persistence of infection [56,130,131]. According to Early et al. (2011), apoptosis leads to the loss of integrity of the vacuole membrane and causes the MAH to rely on Brownian motion to encounter the macrophage cytoplasmic membrane and transfer to the cytoplasm in order to escape from the macrophage [56]. Another possibility is that the MAH in the apoptotic body is not exposed to the extracellular space as the apoptotic body is taken up by secondary macrophages. Because a part of the organism died and only the remaining viable bacteria survived, MAHs taken up by fresh secondary macrophages were expected to be more invasive and pathogenic, and they were expected to cause intracellular replication [56,57]. In a study by Danelishvili et al. (2018), genes that induced rapid apoptosis and entered secondarily infected host cells were identified, and among them, MAVA5_06970, which was postulated to be a secreted protein, was studied [59]. This protein was necessary for virulence and survival in macrophages and mice, and in particular, interacted with ECM proteins and the pro-inflammatory cytokine osteopontin (OPN) [59]. OPN is an important protein involved in the physiological and pathological processes that cause bacterial transmission and disease progression, and the direct correlation between OPN expression levels and clinical outcomes was reported in patients with mycobacterial infection [59,132]. Therefore, it was proposed that MAVA5_06970 binds to OPN to interfere with its function and induce apoptosis in macrophages, consequently acting as an effector by limiting the activation of the type I immune pathway in vivo, and as a result, explaining the mechanisms of cell-to-cell spread of MAH [59] (Table 2, Figure 2).

## 4. Conclusions

MAC, a collection of species representing infectious diseases caused by NTM, are ubiquitous bacteria that can inhabit non-biological or biological resources such as soil, water, food and animals. Despite the MAC subspecies sharing a high genetic identity, they exhibit host specificity with a distinct disease spectrum. In particular, MAC subspecies originating from a common ancestor through phylogenetic analysis are believed to have evolved in a host-specific manner. MAP is tightly closed and has a conservative genome with little mutation, whereas MAH exhibits an open pan-genome with a large gene repertoire. It is believed that MAH has undergone genetic evolution under selective or environmental pressures, allowing adaptation to various host and environmental niches. As discussed in this review, MAHs are able to inhabit non-living resources such as shower heads and PVC pipes by forming biofilms. In addition, surface-exposed proteins can be used to promote adhesion to host epithelial cell membranes or to cross mucosal barriers through skeletal rearrangement of host cells. MAH actively participates in the establishment of an intracellular environment for replication and survival within the macrophage, such as basically avoiding oxidative stress, altering the metabolism in a nutrient-restricted state, or hijacking host proteins to inhibit maturation of the phagosome. Additionally, OMV was used to export effector proteins for early survival of MAH in the macrophage. Cell death of macrophage in MAH infection has not yet been precisely defined. However, the mechanism by which some MAHs escape from dead macrophages and are taken up bystander macrophage may explain the cell-to-cell transmission of MAH. Therefore, MAH has exhibited strategies such as immune regulation and evasion to maintain persistent infection by adapting to the niche, whether it is the environment or the host, rather than destroying the host. This genetic understanding may provide new insight into novel therapeutic approaches targeting pangenesis-associated genes of MAC.

## Figures and Tables

**Figure 1 ijms-22-03011-f001:**
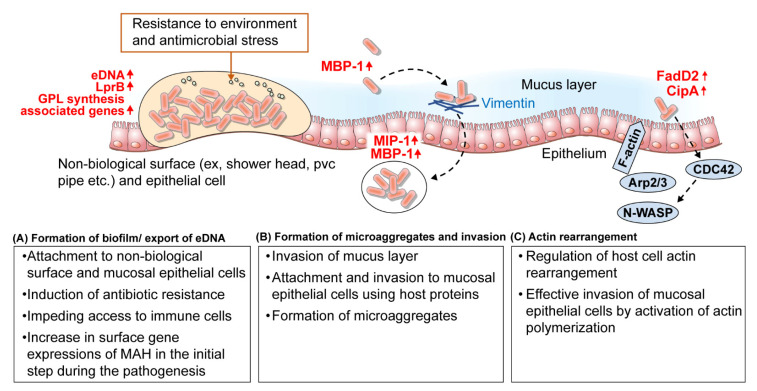
Summary of the strategy of MAH related to mucosal epithelial cell adhesion and invasion. (**A**) Biofilm formation and eDNA export, which greatly contributes to colonization, persistence and drug resistance, are important survival strategies for MAH. MAH lipoprotein (LprB) and GPL-related genes are involved in biofilm formation, and biofilms are involved in adhesion to mucosal epithelium and non-biological surfaces including PVC pipes, shower heads, and hot tubs. (**B**) MAH binds to host proteins such as vimentin, and attaches to epithelial cells, and planktonic bacteria are rapidly recruited to form microaggregates. Furthermore, the MAH microaggregates interact with the host protein filamin A through MIP-1 and further manipulate the host cytoskeleton to efficiently invade the host epithelium. (**C**) MAH can effectively invade mucosal epithelial cells by cytoskeletal rearrangement of host cells by activating the Cdc42 signaling pathway by fadD2, cipA, and other downstream transcriptional regulators. CipA, a domain similar to the PXXP motif of the human piccolo protein; eDNA, extracellular DNA; Fad2, fatty acyl coenzyme A synthase; GPL, glycopeptidolipid; LrpB, leucine-responsive regulatory protein B; MBP-1, microaggregate binding protein 1; MIP-1, microaggregate invasion Protein-1.

**Figure 2 ijms-22-03011-f002:**
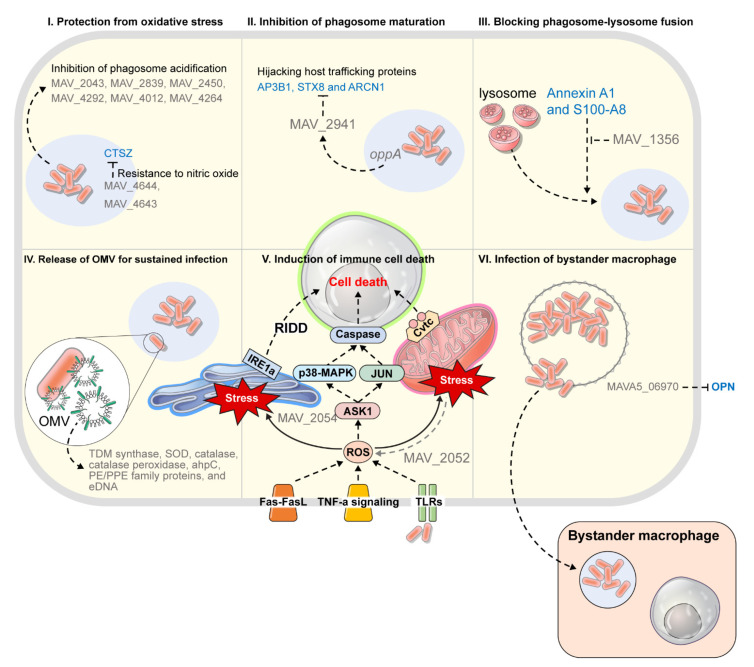
Summary of survival tactics of MAH within phagocytes from host innate immune effectors MAH infection. Six representative survival strategy of MAH for a series of processes after macrophage phagocytosis was presented. **I**. Protection from oxidative stress. In the early stages of macrophage infection, MAH expresses genes resistant to oxidative stress. **II**. Inhibition of phagosome maturation. MAH proteins expressed in the early stages of infection hijack host trafficking proteins, and interfere with their actions, eventually delaying the maturation of the endosome. **III**. Blocking phagosome-lysosome fusion. MAH proteins regulate phagosome membrane or block phagosome-lysosome fusion. **IV**. Release of OMV for sustained infection. MAH releases infection maintenance substances such as TDM synthase, SOD, catalase, catalase peroxidase, ahpC, PE / PPE family protein, and eDNA through OMV. **V**. Induction of immune cell death. MAH-infected macrophages produce TNF, Fas, and TLR-dependent ROS, thereby activating caspase 8 through ASK1/p38 MAPK signals, activating caspase 9 through mitochondrial death signaling, or ER stress-mediated IRE1α-RIDD pathway. And eventually undergo cell death. **VI**. Infection of bystander macrophage. When cell death is induced, the vacuole membrane loses its integrity, and MAH relies on Brownian motion to meet the macrophage cytoplasmic membrane and transfer to the cytoplasm. Eventually, highly invasive MAH can escape macrophages or bacteria remaining in the apoptic body can be taken up by bystander macrophages. ahpC, alkyl hydroperoxide reductase; AP3B1, subunit beta-1 of adaptor-related protein complex 3; ARCN1, Archain 1; ASK1, Apoptosis signal-regulating kinase 1; CTSZ, cathepsin Z; Cytc, cytochrome c; eDNA, extracellular DNA; IRE1a, Inositol-requiring enzyme-1a; JUK, Jun kinase; OMV, outer membrane vesicle; OPN, osteopontin; PE/PPE family proteins, the protein with the motifs Pro-Glu and Pro-Pro-Glu; PI3K, human phosphatidylinositol 3-kinase; p38-MAPK, p38 mitogen-activated protein kinases; RIDD, regulated Ire1-dependent decay; ROS, reactive oxygen species; SOD, superoxide dismutase; STX8, syntaxin-8; S100-A8, S100 calcium-binding protein A8; TDM, trehalose 6,6′-dimycolate; TNF, tumor necrosis factor.

## Data Availability

No new data were created or analyzed in this study. Data sharing is not applicable to this article.

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
