# Peer review of "Genetic Involvement of Mycobacterium avium Complex in the Regulation and Manipulation of Innate Immune Functions of Host Cells"

_ijms, 2021, doi:10.3390/ijms22063011_

Round 1

Reviewer 1 Report

Review

This is a comprehensive literature review entitled “Genetic involvement of Mycobacterium avium complex in the regulation and manipulation of innate immune functions of host cells” The review authored by Min-Kyoung Shin and Sung Jae Shin is well written and the text in clear and easy to read.

The review addresses the need for a better understanding of the pathogenesis of Mycobacterium avium complex (MAC) mycobacteria which belong to Nontuberculous mycobacteria (NTM) and responsible for the most common cause of pulmonary disease worldwide. This review provides an opportunity to bring additional awareness about these neglected mycobacteria. In the first section of this review, the authors provided a brief genomic comparison of different MAC subspecies using genetic features summarized in Table 1 of the manuscript. In the second section, they explored the mechanisms by which MAC induced the disease and how these pathogens escape the immune response.

Below are my major and minor comments:

Major comments:

  1. The growth of NTM or MTB requires the bioavailability of iron and it is well documented that Mycobacteria have key genetic features which are involved in iron-binding siderophore synthesis with the goal to compete efficiently against the host for iron acquisition, a critical mechanism for the virulence of mycobacteria including MAC. Furthermore, iron metabolism disruption in host could also shift host immune response. The authors should add a section or provide information about the genetic features of MAC which are involved in iron acquisition and their importance in the virulence of the mycobacteria MAC.
  2. Although Table 2 is well prepared and detailed, they authors should link the information presented in Table 2 with related sections in the manuscript. Table 2 should be cited at least once in the manuscript. The references in the text and the table should match if both are referring to the same scientific content. For instance, in line 304, the authors cited the reference 64 and 66 while in the table 2 the references 57 and 59 were cited, although both contents were discussing the mechanisms by which MAH adhere and invade host epithelia.

In addition, “Furthermore, McNamara et al. (2012) reported that alkyl hydroperoxide reductase (ahpC, MAV_2839) and isocitrate lyase (aceA, MAV_4682) were uniquely expressed after MAH were exposed to macrophages [89].” The authors cited reference 89 in this section. However, in table 2 Strategy 2, the reference 82 was mentioned. Both contents were similar. The authors should provide some clarifications about these references and check for other potential reference issues.

Minor comments:

  1. Line 276: full stop after [57, 58]
  2. Line 277: full stop after [56]
  3. Line 279: full stop after [59-60]
  4. Line 385: correct anti-biotics with antibiotics
  5. From line 397 to line 408: check the line spacing
  6. Line 494: Add a space after [97, 103]
  7. Line 555. In the figure 2, the authors should check what they are referring to as A in the caption. The section B was not mentioned in the caption or in the figure.
  8. Line 573: Add a full stop after the ...3’ kinase.
  9. In the caption of Fig 2, the authors should list the abbreviation details in an alphabetic order.
  10. Lines 587 and 594, the authors should move the synonym LL-37 of cathelicidin to line 587
  11. The figure captions should have a reduced font size in comparison to the text itself
  12. The page number sequences should be checked.

Author Response

Reviewer #1

This is a comprehensive literature review entitled “Genetic involvement of Mycobacterium avium complex in the regulation and manipulation of innate immune functions of host cells” The review authored by Min-Kyoung Shin and Sung Jae Shin is well written and the text in clear and easy to read.

The review addresses the need for a better understanding of the pathogenesis of Mycobacterium avium complex (MAC) mycobacteria which belong to Nontuberculous mycobacteria (NTM) and responsible for the most common cause of pulmonary disease worldwide. This review provides an opportunity to bring additional awareness about these neglected mycobacteria. In the first section of this review, the authors provided a brief genomic comparison of different MAC subspecies using genetic features summarized in Table 1 of the manuscript. In the second section, they explored the mechanisms by which MAC induced the disease and how these pathogens escape the immune response.

Answer: We appreciate the reviewer’s clear summary of our work. We also appreciate the reviewer for the generous comments and recognition of the hypothesis-generating nature of our work.

Below are my major and minor comments:

 Major comments:

  1. The growth of NTM or MTB requires the bioavailability of iron and it is well documented that Mycobacteria have key genetic features which are involved in iron-binding siderophore synthesis with the goal to compete efficiently against the host for iron acquisition, a critical mechanism for the virulence of mycobacteria including MAC. Furthermore, iron metabolism disruption in host could also shift host immune response. The authors should add a section or provide information about the genetic features of MAC which are involved in iron acquisition and their importance in the virulence of the mycobacteria MAC.

Answer: We appreciated valuable comments on our manuscript. As reviewer commented, iron acquisition is a very important trait in mycobacteria for survival in the host. From section 3 of this manuscript, we have focused on the genetically diverse MAH. In particular, we have listed and described genes whose functions and clear mechanisms that can survive host defense strategies have been discovered. In MAH, as far as we know, there is no reference directly explaining the role of iron metabolism, thus, we have described the role of iron metabolism in MAP genetically close to MAH among MAC in Section 2.2 (lines 215-234).

  1. Although Table 2 is well prepared and detailed, they authors should link the information presented in Table 2 with related sections in the manuscript. Table 2 should be cited at least once in the manuscript. The references in the text and the table should match if both are referring to the same scientific content. For instance, in line 304, the authors cited the reference 64 and 66 while in the table 2 the references 57 and 59 were cited, although both contents were discussing the mechanisms by which MAH adhere and invade host epithelia.

Answer: Thank you for the critical comments. We completely agree on your suggestions. We have replaced these references pointed out with the correct references,and rechecked in the revised manuscript.

In addition, “Furthermore, McNamara et al. (2012) reported that alkyl hydroperoxide reductase (ahpC, MAV_2839) and isocitrate lyase (aceA, MAV_4682) were uniquely expressed after MAH were exposed to macrophages [89].” The authors cited reference 89 in this section. However, in table 2 Strategy 2, the reference 82 was mentioned. Both contents were similar. The authors should provide some clarifications about these references and check for other potential reference issues.

Answer: We appreciated the suggestion made by the reviewer. We replaced ref.85 with the correct one, as the reviewer pointed out, and rechecked the references throughout the manuscript. (line 461)

Minor comments:

1. Line 276: full stop after [57, 58]

Answer: Thank you for the careful point of the reviewer. We have added a full stop after [57, 58] (line 295).

2. Line 277: full stop after [56]

Answer: Thank you for the careful point of the reviewer. We have added a full stop after [56] (line 296).

3. Line 279: full stop after [59-60]

Answer: Thank you for the careful point of the reviewer. We have added a full stop after [56, 59-60] (line 298).

4. Line 385: correct anti-biotics with antibiotics

Answer: Thank you for the careful point of the reviewer. We corrected anti-biotics with antibiotics (line 406).

5. From line 397 to line 408: check the line spacing

Answer: Thank you for the careful point of the reviewer. We checked check the line spacing (line 417-430).

6. Line 494: Add a space after [97, 103]

Answer: Thank you for the careful point of the reviewer. We added a space after [97, 103] (line 518).

7. Line 555. In the figure 2, the authors should check what they are referring to as A in the caption. The section B was not mentioned in the caption or in the figure.

Answer: Thank you for the careful point of the reviewer. We deleted (A) from the figure 2 caption (line 609).

8. Line 573: Add a full stop after the ...3’ kinase.

Answer: Thank you for the careful point of the reviewer. We have added a full stop after the corresponding sentence (line 621).

9. In the caption of Fig 2, the authors should list the abbreviation details in an alphabetic order.

Answer: Following the reviewer's comment, we listed the abbreviation details in alphabetical order in the caption in Figure 2.

10. Lines 587 and 594, the authors should move the synonym LL-37 of cathelicidin to line 587

Answer: Following the reviewer's comment, we moved the cathelicidin synonym LL-37 to line 584.

11. The figure captions should have a reduced font size in comparison to the text itself

Answer: According to the reviewer's comment, the font size of the figure caption has been revised to be smaller than the text.

12. The page number sequences should be checked.

Answer: Thank you for your critical pointing. We corrected to sequential page numbers. 

Reviewer 2 Report

The manuscript reviews the genetic properties of Mycobacterium avium complex, and in particular the subsp. hominissuis, in the context of the host-pathogen interplay. Gaining insight into the bacteria regulation of the host immunity should help to develop novel therapies. Nontuberculous mycobacteria, and in particular M. avium, can be abundant in the environment and cause pulmonary infection. Most vulnerable population, as immunocompromised patients or children with cystic fibrosis, are prone to be infected. However, there is still not sufficient data to explain the distinct subspecies dissemination.

The authors have a long and solid background on the genetic characterization of M. avium mycobacteria specie and the review is well documented. A lot of effort has been applied to prepare summary tables and pictures. In particular it includes two tables that provide a lot of information on the distinct M. avium subspecies. The paper should be of general interest for the scientific community. However, before publishing some part of the text should be revised and polished. In addition, a greater effort in the data presentation would be helpful for the reader.

Point-by-point comments:

  • Table 2 should be cited beforehand in the text. No mention to table 2 is found.
  • Position of table 2 is not appropriate, as no previous information on the data within the table is given. Besides, some columns in table 2 give some redundant information in some sections. For example, in strategy 2, the same subtitle is provided in the 1st, 4th and 5th I think it is better to include more specific information in the5th column (as done for example in strategy 4 section).
  • Figure 2 legend is too long. It should be more schematic. The other information can be shifted to the main text. Also, it has an A) section that is not found in the picture, and no other sections are indicated. Besides, the abbreviation list should start after a fullstop.
  • Revise, figure 2 abbreviation list. For example TDM is not trehalose, should be trehalose dymicolate
  • The manuscript has some sections that should preferably be revised for clarity. Mostly from section 3.2 onwards there are some sentences too long and sometimes even confusing. For example, first and second paragraphs of section 3.2 should be rephrased.
  • Check also sentence lines 591-593: it says polymyxin is a surrogate of the antimicrobial peptide, but no peptide has been mentioned before.
  • Line 606: pathogenicity here does not seem to be correct. Maybe, it can be substituted with “resistance”.
  • Also some information in these last sections is a bit disorganized.
  • It is also essential to revise the references. For example, ref. 115 does not seem to talk about LL-37 and polymyxin, or some other antimicrobial peptide.
  • Preferably conclusion section should be more specific, showing the main conclusions from the data provided in the review.
  • Minor points:

Page 2, line 193: add full name for SGM abbreviation

Better substitute anti-microbial with antimicrobial

Also better write antibiotics, not anti-biotics

Some species are not indicated in italics

To facilitate reading, check that full names for all abbreviations are provided

Author Response

Reviewer #2

The manuscript reviews the genetic properties of Mycobacterium avium complex, and in particular the subsp. hominissuis, in the context of the host-pathogen interplay. Gaining insight into the bacteria regulation of the host immunity should help to develop novel therapies. Nontuberculous mycobacteria, and in particular M. avium, can be abundant in the environment and cause pulmonary infection. Most vulnerable population, as immunocompromised patients or children with cystic fibrosis, are prone to be infected. However, there is still not sufficient data to explain the distinct subspecies dissemination.

The authors have a long and solid background on the genetic characterization of M. avium mycobacteria specie and the review is well documented. A lot of effort has been applied to prepare summary tables and pictures. In particular it includes two tables that provide a lot of information on the distinct M. avium subspecies. The paper should be of general interest for the scientific community. However, before publishing some part of the text should be revised and polished. In addition, a greater effort in the data presentation would be helpful for the reader.

Answer: We thank you for dedicating valuable time and providing expert comments on our manuscript. We also appreciate the reviewer for the generous comments and recognition of the hypothesis-generating nature of our work. We have carefully considered your comments and suggestions, as well as those offered by the reviewers. We made our best effort to address all of the concerns raised by the reviewer. As advised by the reviewer, we have revised the tables and figures that provide information for this review. Sections and paragraphs pointed out by reviewer have been rewritten to avoid confusion and ambiguity. In addition, we replaced these references pointed out by the reviewer with the correct ones, and rechecked those throughout the manuscript. We hope that our revisions have improved the clarity of our manuscript and allayed any concerns regarding the quality of our work. We have individually addressed the reviewer’s comments and have highlighted the revised part in the manuscript.

Point-by-point comments:

  • Table 2 should be cited beforehand in the text. No mention to table 2 is found.

Answer: Thank you for the critical comment. We have cited Table 2 accordingly in the revised manuscript.

  • Position of table 2 is not appropriate, as no previous information on the data within the table is given. Besides, some columns in table 2 give some redundant information in some sections. For example, in strategy 2, the same subtitle is provided in the 1st, 4thand 5th I think it is better to include more specific information in the 5th column (as done for example in strategy 4 section).

Answer: Thank you for the valuable comments. First, we have placed Table2 in the middle of the manuscript following your suggestion (pages 10-14). Second, we agree with the reviewer's opinion on the insufficeint uniformity of each column in Table 2. As suggested, the information form of the column in Table 2 has been completely reconstructed. We indicated all changes in a blue font.

  • Figure 2 legend is too long. It should be more schematic. The other information can be shifted to the main text. Also, it has an A) section that is not found in the picture, and no other sections are indicated. Besides, the abbreviation list should start after a full stop.

Answer: We appreciate the reviewer for the excellent comments. We deleted (A) from the figure 2 caption and added a full stop before starting the abbreviation list. In addition, following the reviewer's suggestion, the Figure 2 legend has been revised schematically (line 609-626).

  • Revise, figure 2 abbreviation list. For example TDM is not trehalose, should be trehalose dymicolate

Answer: Thank you for the critical comments. In the caption of Figure 2, the details of the abbreviations have been checked and revised (line621-626).

  • The manuscript has some sections that should preferably be revised for clarity. Mostly from section 3.2 onwards there are some sentences too long and sometimes even confusing. For example, first and second paragraphs of section 3.2 should be rephrased.

Answer: Thanks for your valuable suggestions. We have rewritten lengthy sentences in a brief manner in section 3.2. We indicated all changes in a blue font. (line 436-441, 445-449, 513-518)

  • Check also sentence lines 591-593: it says polymyxin is a surrogate of the antimicrobial peptide, but no peptide has been mentioned before.

Answer: Thank you for the critical comments. We agree with the opinion of reviewer, and we revised the pointed out to “a surrogate for the host antimicrobial peptide”, and this paragraph seems to be confusing, so we have rewritten the sentence for more clarity. We indicated all changes in blue font. (lines 588-593)

  • Line 606: pathogenicity here does not seem to be correct. Maybe, it can be substituted with “resistance”.

Answer: We appreciated your suggestion. We have replaced the term of “sensitivity and pathogenicity” to “resistance” according to your suggestion (line 603-604).

  • Also some information in these last sections is a bit disorganized.

Answer: Thank you for the critical comments. We reconfirmed the last session according to the logical flow of the manuscript, and re-wrote any confused or ambiguous parts. We indicated all changes in a blue font (lines 630-639, 653-658, 669-672).  

  • It is also essential to revise the references. For example, ref. 115 does not seem to talk about LL-37 and polymyxin, or some other antimicrobial peptide.

Answer: We appreciated the suggestion made by the reviewer. We replaced ref.115 with the correct ref, as the reviewer pointed out, and rechecked that other references were also correct (line 591).

  • Preferably conclusion section should be more specific, showing the main conclusions from the data provided in the review.

Answer: Thank you for the careful point of the reviewer. We have specifically described the conclusion section as advised by the reviewer.

  • Minor points:

Page 2, line 193: add full name for SGM abbreviation

Answer: Thank you for the careful point of the reviewer. We described fill name of SGM. (line 194)

Better substitute anti-microbial with antimicrobial

Answer: Thank you for the careful point of the reviewer. We have revised anti-microbial to antimicrobial throughout the manuscript.

Also better write antibiotics, not anti-biotics

Answer: Thank you for the careful point of the reviewer. We corrected anti-biotics with antibiotics throughout the manuscript.

Some species are not indicated in italics

Answer: Thank you for the critical point of the reviewer. We have changed bacteria species in italic throughout the manuscript. We indicated all changes in a blue font.

To facilitate reading, check that full names for all abbreviations are provided

Answer: Thank you for the careful point of the reviewer. We have checked that the full name of the abbreviation has been provided throughout the text.

Round 2

Reviewer 1 Report

All my concerns have been addressed in the revised version of the manuscript.

Reviewer 2 Report

All comments have been addressed.